# 500 microkelvin nanoelectronics

Matthew Sarsby[1,3], Nikolai Yurttagül[1,3] & Attila Geresdi [1,2 ✉]

Fragile quantum effects such as single electron charging in quantum dots or macroscopic coherent tunneling in superconducting junctions are the basis of modern quantum technologies. These phenomena can only be observed in devices where the characteristic spacing between energy levels exceeds the thermal energy, $k_BT$, demanding effective refrigeration techniques for nanoscale electronic devices. Commercially available dilution refrigerators have enabled typical electron temperatures in the 10 to 100 mK regime, however indirect cooling of nanodevices becomes inefficient due to stray radiofrequency heating and weak thermal coupling of electrons to the device substrate. Here, we report on passing the millikelvin barrier for a nanoelectronic device. Using a combination of on-chip and off-chip nuclear refrigeration, we reach an ultimate electron temperature of $T_e = 421 \pm 35$ µK and a hold time exceeding 85 h below 700 µK measured by a self-calibrated Coulomb-blockade thermometer.

[1] QuTech and Kavli Institute of Nanoscience, Delft University of Technology, 2600 GA Delft, The Netherlands. [2] Present address: Department of Microtechnology and Nanoscience, Chalmers University of Technology, SE 41296 Gothenburg, Sweden. [3] These authors contributed equally: Matthew Sarsby, Nikolai Yurttagül. ✉ email: geresdi@chalmers.se

Accessing the microkelvin regime[1] holds the potential of enabling the observation of exotic electronic states, such as topological ordering[2], electron-nuclear ferromagnets[3,4], p-wave superconductivity[5], or non-Abelian anyons[6] in the fractional quantum Hall regime[7]. In addition, the error rate of various quantum devices, including single-electron charge pumps[8] and superconducting quantum circuits[9], could improve by more effective thermalization of the charge carriers.

The conventional means of cooling nanoelectronic devices relies on the heat flow $\dot{Q}$ between the refrigerator and the conduction electrons, mediated by phonon–phonon coupling in the insulating substrate, $\dot{Q}_{p-p} \propto T_{p1}^4 - T_{p2}^4$ with $T_{p1}$ and $T_{p2}$ being the respective phonon temperatures, and electron–phonon coupling in the device $\dot{Q}_{e-p} \propto T_e^5 - T_p^5$ where $T_e$ is the electron temperature[10,11], both heat flows rapidly diminishing at low temperatures. In a typical dilution refrigerator, $T_p$ is larger than 5 mK, and specially built systems reach 1.8 mK[12], which limits $T_e$ larger than $T_p$ to be in the millikelvin regime. The lowest static electron temperature reached with this technique was $T_e = 3.9$ mK[13] in an all-metallic nanostructure, and other experiments reached similar values[14–16] in semiconductor heterostructures.

This technological limitation can be bypassed by adiabatic magnetic refrigeration, which relies on the constant occupation probability of the energy levels of a spin system, proportional to $\exp(-g\mu Bm/k_B T)$ in the absence of heat exchange with the environment[17]. Here, $k_B T$ is the thermal energy at a temperature of $T$, $g\mu B$ is the energy split between adjacent levels at a magnetic field of $B$, and $m$ is the spin index. Thus, the constant ratio $B/T$ allows for controlling the temperature of the spin system by changing the magnetic field. Exploiting the spin of the nuclei[18,19], this technique has been utilized to cool nuclear spins in bulk metals down to the temperature range of $T$ around 100 pK[20]. If only Zeeman splitting is present, which is linear in $B$, the ultimate temperature is limited by the decreasing molar heat capacity, $C_n = \alpha B^2/T^2$ in the presence of finite heat leaks, $\dot{Q}_{leak}$. Here, the prefactor $\alpha = N_0 I(I+1)\mu_n^2 g_n^2/3k_B$ contains $I$, the size of the spin; $g_n$, the g-factor; $N_0$, the Avogadro number; and $\mu_n$, the nuclear magneton.

Bulk nuclear cooling stages have predominantly been built of copper[19], owing to its high thermal conductivity, beneficial metallurgic properties, and weakly coupled nuclear spins, which allows for magnetic refrigeration of the nuclear spins down to 50 nK[21]; however, the weak hyperfine interaction results in a decoupling of the electron system at much higher temperatures, around 1 μK in bulk samples. Another material, a Van Vleck paramagnet, $PrNi_5$, has been used as a bulk nuclear refrigerant exploiting its interaction-enhanced heat capacity in a temperature range above 200 μK[22] even in dry dilution refrigerators[23].

Integrating the nuclear refrigerant with the nanoelectronic device yields a direct heat transfer between the electrons and nuclei $\dot{Q}_{e-n} = \alpha\kappa^{-1}B^2(T_e/T_n - 1)$ per mole. Here, $\kappa$ is the Korringa constant, and $T_e$ and $T_n$ are the electron and nuclear spin temperatures, respectively. Owing to an $\alpha/\kappa$ ratio 60 times better than that of copper, indium has recently been demonstrated as a viable on-chip nuclear refrigerant[24]. In addition, indium allows for on-chip integration of patterned thick films by electrodeposition and has been demonstrated as a versatile interconnect material for superconducting quantum circuits[25].

However, on-chip nuclear refrigeration is limited by the large molar heat leaks $\dot{Q}_{leak}$, which thus far limited the attainable electron temperatures above 3 mK both for copper ($\dot{Q}_{leak} = 0.91$ pW mol$^{-1}$ (ref. [26])) and indium ($\dot{Q}_{leak} = 0.69$ pW mol$^{-1}$ (ref. [24])) as on-chip refrigerant, with a total heat leak of approximately 4 pW in both of these experiments. Furthermore, only very short hold times were attained with the chips warming up during the magnetic field sweep,

limiting the practical applications of these devices. In contrast, encapsulating the chip in a microkelvin environment has led to superior cooling performance using copper, yielding $T_e = 2.8$ mK with a hold time of approximately 1 h below 3 mK[27].

Here, we report on a combined on- and off-chip nuclear demagnetization approach, which surpasses the 1 mK barrier in a nanoelectronic device. We demonstrate an electron temperature below 500 μK by combining on-chip indium cooling fins with an off-chip parallel network of bulk indium leads attached to a copper frame. By demagnetization cooling of all components together, we greatly reduce the heat leak to the nanoscale device, enabling both a record low final electron temperature and long hold times in excess of 85 h, demonstrating the viable utilization for quantum transport experiments in the microkelvin regime.

## Results

**Setup for on- and off-chip indium nuclear demagnetization.** We discuss the experimental setup following the cross-sectional drawing in Fig. 1a. Our design is accommodated by a commercial dilution refrigerator (MNK126-700; Leiden Cryogenics) with several microwave-tight radiation shields to reduce electronic noise coupling to the nanostructure. The magnetic field required for magnetic cooling was applied by a superconducting solenoid with a rated field of 13 T. The device was installed in the field center; however, parts of the off-chip refrigerants were subjected to smaller magnetic fields defined by the field profile of the solenoid, which we account for by listing the effective molar amount of the materials.

We attach the copper frame (1.7 mol total, 0.32 mol effective amount) to the mixing chamber via an aluminum-foil heat switch[28], which is activated by a small solenoid at $\approx 10$ mT of applied magnetic field. Crucially for our design, the electrical measurement lines are not connected to the copper stage, rather to the parallel network of nuclear cooled indium wires with a 23 mmol total and 17 mmol effective amount per line. These wires are attached to the copper frame by an epoxy resin membrane (nominal thickness of 40 μm, Fig. 1b) which provides an additional layer of thermal isolation during nuclear cooling. The electrical and thermal contact to the device is made by press welding the indium wires onto electroplated indium bonding pads on the chip (Fig. 1c).

We apply both electronic filtering and microwave shielding in order to reach sub-1 mK electron temperatures. We thermalize all electrical lines by custom-made copper powder filters[29] at each stage of the dilution refrigerator and a fifth-order RC filter with a cutoff frequency of 50 kHz at the mixing chamber. To decouple the cold chip below 1 mK from the thermal noise of the mixing chamber[30], we install an additional set of copper powder filters within the copper stage which is demagnetized together with the indium lines (Fig. 1d, e). Further details on the measurement circuit are listed in Supplementary Figs. 1–4 and in the Methods.

**Integrated Coulomb-blockade thermometry.** To directly measure the electron temperature, we utilize a Coulomb-blockade thermometer (CBT)[31]. CBTs rely on the universal suppression of charge fluctuations in small metallic islands enclosed between tunnel barriers at low electron temperatures. With a total capacitance of each island, $C_\Sigma$, we define the charging energy for a tunnel junction array of length $N$ to be $E_C = e^2/C_\Sigma \times (N-1)/N$ with $e$ being the charge of a single electron. In the weak charging regime, $k_B T_e \gg E_C$, the full-width at half-maximum of the charging curve is $eV_{1/2} = 5.439Nk_B T$, providing the possibility of primary thermometry independent of $E_C$ defined by the device geometry[32]. Crucially for temperature measurements during

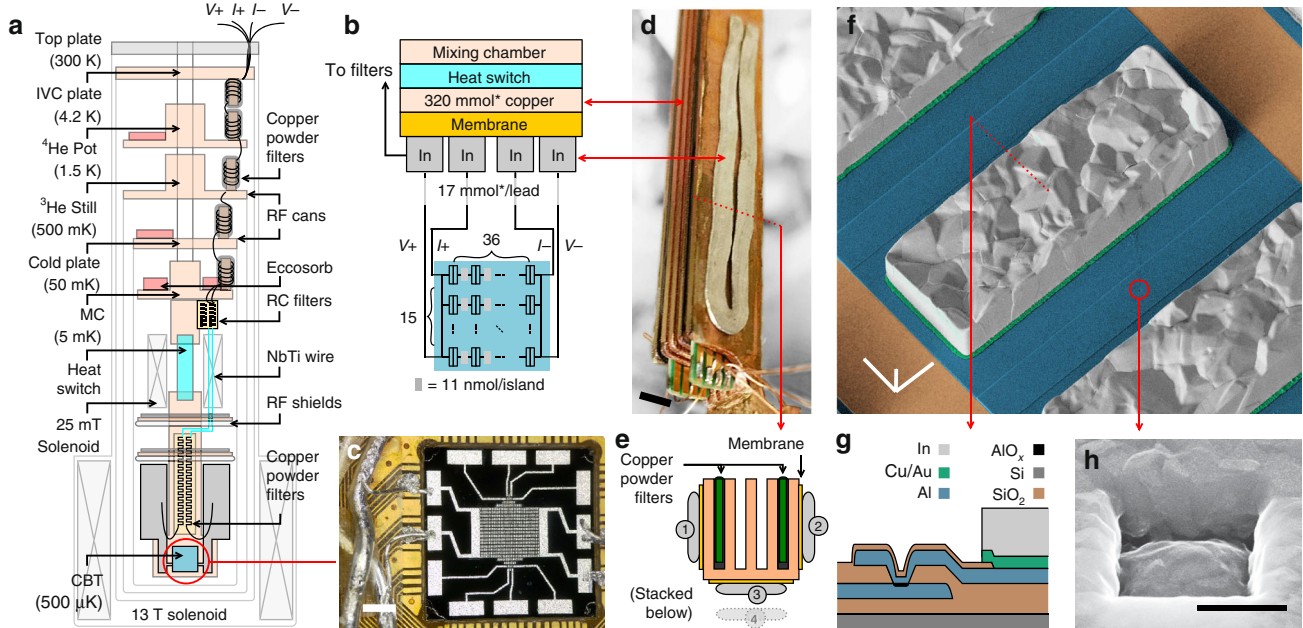

**Fig. 1 The experimental setup for microkelvin nanoelectronics. a** The cross-sectional sketch of the setup integrated into a dilution refrigerator with the characteristic temperatures listed on the left. The device, a Coulomb-blockade thermometer (CBT) with the integrated indium refrigerant, is shown as a cyan box, encircled in red. **b** The thermal path between the mixing chamber of the dilution refrigerator and the CBT (highlighted in cyan, the gray blocks denote In cooling fins). The electrical signal lines pass through the indium blocks. The effective molar amount of the refrigerant material is listed in each block, respectively. **c** Photograph of the CBT chip with the press-welded connections to the off-chip indium stages consisting of 1 mm diameter indium wires are visible on the left. **d** Photograph of the indium refrigerant mounted on the copper block with an electrically insulating membrane in between. The picture was taken during the assembly of the stage. The slots of the copper block contain the copper powder filters of the electrical lines to reduce radiofrequency heating, see also the cross-sectional image in panel **e**. **f** False-colored scanning electron micrograph of an island of the CBT, with a cross-sectional schematics shown in panel **g** along with the color legend. The encircled $Al/AlO_x/Al$ tunnel junction is shown in higher magnification in panel **h**. The scale bars denote 2 mm (**c**), 5 mm (**d**), 20 μm (**f**), and 500 nm (**h**), respectively. See the Data availability section for raw images.

demagnetization cooling, CBTs exhibit no sensitivity to the applied magnetic field[33].

We fabricated a $36 \times 15$ array with an ex situ $Al/AlO_x/Al$ tunnel junction process[24,34] to create junctions of $780 \times 780$ nm$^2$ in size (Fig. 1h). Since the highly resistive tunnel junctions with $R_j = 35$ kΩ correspond to a high thermal barrier, we electro-deposited indium cooling fins ($50 \times 140 \times 25.4$ μm$^3$ corresponding to 11.4 nmol, see Fig. 1f, g) on each island for local nuclear cooling. We note however that the measured device has only five conducting lines, resulting in an $N \times M = 36 \times 5$ CBT array. The details of the device fabrication are previously described in ref. [24].

In Fig. 2a, we show the normalized differential conductance $G(V)/G_t$ of the CBT at different temperatures obtained by a conventional low-frequency lock-in technique at 19.3 Hz. We use the master equation of single electron tunneling[31,32] to find the electron temperature of each curve and the charging energy as a global fit parameter, $E_C = 232.6 \pm 0.8$ neV $= k_B \cdot 2.7$ mK, yielding $C_\Sigma = 670 \pm 2$ fF. Remarkably, at a small magnetic field of $B = 45$ mT required to suppress superconductivity, we achieve an electron temperature $T_e = 7.07 \pm 0.09$ mK by phonon cooling, attesting to the well-shielded environment of the device. At this low temperature, we had to account for the Joule heating of the CBT chip[35], see Supplementary Note 1 and Supplementary Fig. 6. Upon calibration, we use the zero bias conductance $G(V = 0)$ for continuous thermometry, see Supplementary Note 1. We note that we observe a temperature-independent $\Delta G_t/G_t \approx 0.03$ magnetoresistance between zero field and 12 T (inset of Fig. 2a), which we account for during the magnetic field ramps. To demonstrate the limitations of phonon cooling, we evaluate $T_e$ as a function of the mixing chamber temperature and observe a saturation behavior below 10 mK (Fig. 2b).

In the low-temperature regime, where $E_C$ is larger than $k_B T_e$, random island charge offsets influence the conductance of the CBT[36], which causes an a statistical uncertainty of the inferred $T_e$. To estimate this confidence interval, we use the following numerical procedure. First, we calculate $G/G_t$ by a Markov chain electron counting model with a random set of offset charges for each island. Using the Markov chain Monte Carlo (MCMC) approach, we evaluate the conductances of the tunnel junction chain for 1000 gate charge configurations at a given $T_e$. Finally, we build a histogram of the total CBT conductance, which is the a sum of $M = 5$ randomly selected line conductances. The conclusion of our calculations is that the conductance distribution remains surprisingly narrow for our array consisting of 35 islands in series and 5 rows in parallel and enables a temperature measurement with a $3\sigma$ relative confidence interval of less than 10% for $T_e$ above 400 μK $= 0.15 E_C$. We plot the calculated calibration curve $T_e(G/G_t)$ (blue solid line) together with a $3\sigma$ confidence interval in Fig. 2c (blue shade), which provides the quoted statistical error of $T_e$. We also display the $T_e = 0.4 E_C/k_B = 1.1$ mK limit of the universal behavior[36], where offset charges do not play a role in the device conductance. In addition, we plot (orange curve) the theoretical maximum of $T_e(G/G_t)$, which occurs when all islands are in full Coulomb blockade. For the MCMC calculation, we assumed uniform tunnel junction resistances and island capacitances. Our fabrication process yields a junction resistance variation of $\delta R/R$ less than 10%[24], which yields an upper bound of systematic temperature error of $\Delta T_e/T_e = k(\delta R/R)^2$ around 1%, where $k$ is a geometry-dependent prefactor of the order of unity[37]. The details of our numerical simulations can be found in Supplementary Note 1.

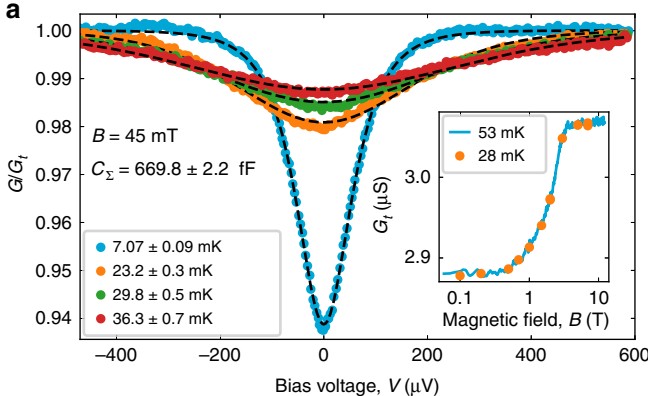

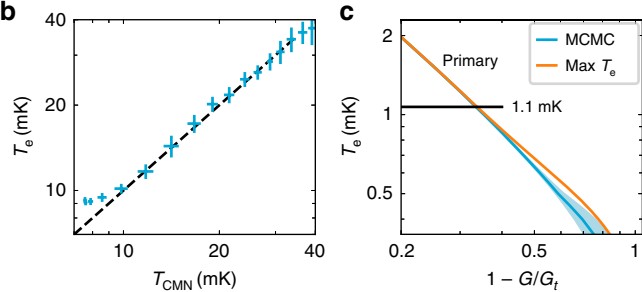

**Fig. 2 Calibration and primary operation of the CBT. a** The voltage-dependent differential conductance, $G$ of the device at different temperatures. The conductance is normalized with the conductance at large bias, $G_t$ and fitted against the single electron tunneling model to determine the electron temperature for each curve and the total capacitance per island, $C_\Sigma = 669.8 \pm 2.2$ fF. The inset shows $G_t$ as a function of the magnetic field, at two distinct temperatures. **b** The electron temperature, $T_e$ at $B = 100$ mT with the heat switch closed as a function of the mixing chamber temperature of the dilution refrigerator, measured by a calibrated cerium magnesium nitrate (CMN) mutual inductance thermometer. The temperature errors are shown as horizontal and vertical lines, respectively. **c** The theoretical zero bias calibration curve based on $E_C$ with the $k_B T_e = 0.4 E_C$ limit[36] depicted as the horizontal line, see text. The theoretical maximum of $T_e(G/G_t)$ with all CBT islands in full Coulomb blockade is denoted by the orange line. The average $T_e$ and the $3\sigma$ confidence interval are shown as the blue solid line and the blue shaded region, respectively, based on the Markov chain Monte Carlo (MCMC) method, see section Integrated Coulomb-blockade thermometry. See the Data Availability section for raw data.

**Electron cooling by nuclear demagnetization.** We initialize a nuclear demagnetization experiment by precooling the device with a closed heat switch at $B_i = 12$ T and reach $T_{e,i} = 13$ mK after precooling for 164 h. After opening the heat switch, we remove the magnetic field with a decreasing rate of $\dot{B}(B) = [-2 \times 10^{-4} \text{ s}^{-1}] \cdot B$ to $B_f = 0.1$ T, which results in a decreasing $T_e$ (Fig. 3a). After finishing the ramp, we find the lowest measured electron temperature $T_e = 421 \pm 35 \text{ μK}$ calibrated using our MCMC model, averaged over a period of 1 h (see inset of Fig. 3a). The quoted $3\sigma$ confidence interval is based on the conductance variation calculated by the MCMC model (see Fig. 2c). We note that the full blockade limit of the CBT also yields $T_{e,\text{min}}$ less than 500 μK (orange line).

After reaching the minimum temperature, the device gradually warms up. We find a warm-up rate of $\dot{T}_e = 1.7 \text{ μK h}^{-1}$, and a hold time of 85 h with $T_e < 700$ μK. This duration was limited by the periodic cryogenic liquid transfer to the host cryostat, which

induced mechanical vibrations and consequently a rapid warmup of the device.

As a proof of concept quantum transport experiment in the microkelvin regime, we measure the finite bias charging curve of the CBT at two different temperatures below 1 mK (circles in Fig. 3c, d). Remarkably, we find an excellent agreement with the calculated curves, using $T_e$ as the single fit parameter (dashed lines). In addition, we observe no distortion of the lineshape due to Joule heating at finite voltage biases, further demonstrating the role of strong electron–nucleus coupling in indium. To demonstrate the wide range of primary thermometry in our regime of interest, we display a set of experimental data taken between $T_e = 480 \text{ μK}$ and 23.2 mK (Fig. 3e) together with the fitted theoretical curves on a logarithmic scale (circles and dashed lines, respectively).

**Internal magnetic field and heat leak calculation.** Next, we turn to the refrigeration properties of indium, and display a series of $T_e(B)$ curves in Fig. 4. Notably, $T_e^2(B^2)$ follows a linear relation with a negative intercept, $-b^2 = -(295 \pm 7 \text{ mT})^2$ (Fig. 4a), which demonstrates that indium does not follow a constant $B/T$ ratio. We attribute our results to the strong quadrupolar splitting due to the inhomogeneous electric field in the crystalline lattice[38]. A resulting effective magnetic field $b$ then adds to the applied magnetic field, $B_{\text{tot}}^2 = B^2 + b^2$ (ref. [38]), which limits the lowest attainable temperature, even if $B \ll b$ (Fig. 4b). Our measurements yield similar $b$ values to earlier nuclear demagnetization experiments in bulk indium samples[39]. This correspondence directly confirms that the measured $T_e$ and the inferred nuclear spin temperature, $T_n$, are close to each other, which is the key requirement for nuclear demagnetization cooling of nanoelectronics.

Based on the obtained internal magnetic field, we estimate the heat leak of the CBT using the warm-up rate $dT_n/dt = \dot{Q}_{\text{leak}}/C_n$, where $C_n = n\alpha B_{\text{tot}}^2/T_n^2$ is the nuclear spin heat capacity with $n$ being the molar amount of the refrigerant. These equations yield $\dot{Q}_{\text{leak}} = -n\alpha B_{\text{tot}}^2 \frac{d(1/T_e)}{dt}$, where $d(1/T_e)/dt$ is the inverse warm-up rate (Fig. 3b) in the limit of $T_n = T_e$. At $B = 100$ mT, we find $\dot{Q}_{\text{leak}} = 26.7$ aW per island, which is lower than the $\dot{Q}_{\text{leak}} = 32$ aW per island obtained earlier using copper[27].

The small heat leak is further attested by the direct demonstration of the adiabaticity of the experiment. In the inset of Fig. 4b, we show a measurement run starting with $B_i = 1$ T and $T_{e,i} = 1.57 \pm 0.01$ mK. After ramping up the magnetic field up to 8 T and back to $B_f = 1$ T, we acquire $T_{e,f} = 1.57 \pm 0.06$ mK. With $B_f = B_i$, the adiabatic efficiency[27] reads $\eta = T_{e,i}/T_{e,f} = 1.00 \pm 0.04$, unity within error margin. This remarkable result demonstrates the outstanding control of $T_e$ by setting the magnetic field, which is unprecedented for nuclear magnetic cooling stages built for nanoelectronics.

**Discussion**

In conclusion, we demonstrated nuclear magnetic cooling of electrons in a nanostructure down to an ultimate temperature of $T_e = 421 \pm 35 \text{ μK}$. Integrating on- and off-chip refrigeration utilizing the strong electron-nucleus coupling of indium, we achieve 85 h of hold time in the microkelvin range, which opens up this, so far unexplored, temperature regime for nanoelectronic devices and quantum transport experiments in pursuit of topological and interacting electron systems. We note that other transition metals with a large $\alpha/\kappa$ ratio[38] can potentially be used in future experiments to target an ultimate temperature regime of $T_e \approx T_n \sim \mu_n g_n b/k_B$ determined by the internal magnetic field $b$. However, comparing our results with earlier experiments featuring on-chip-only refrigeration[24,26]

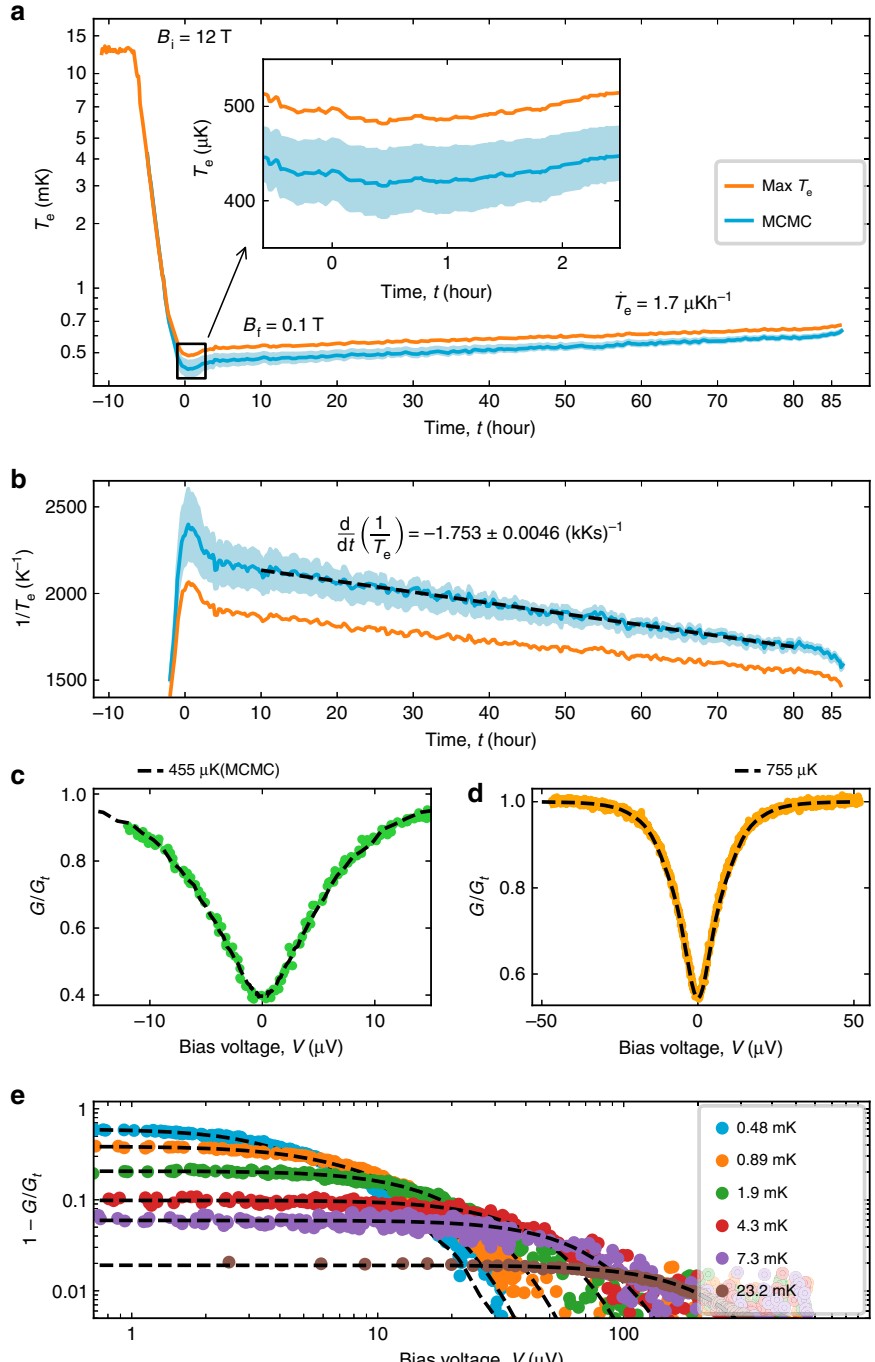

**Fig. 3 Nuclear demagnetization of the CBT to the microkelvin regime. a** A nuclear demagnetization experiment starting from $T_e = 13$ mK and $B_i = 12$ T ramping down to $B_f = 0.1$ T with a ramp rate of $\dot{B}(B) = [-2 \times 10^{-4}\,\text{s}^{-1}] \cdot B$. The inset shows a closer view of the minimum electron temperature reaching well below $T_e = 500$ μK for all calibrations, see text. **b** The inverse temperature $1/T_e$ as a function of time, which is used to estimate the heat leak, see text. Panels **c**, **d** display finite bias voltage data in demonstration of the primary CBT operation in the microkelvin regime. **e** Finite voltage bias data of the CBT in a broad temperature range. In panels **c**–**e**, the experimental data are shown as circles. The dashed lines are the fitted theoretical curves based on the master equation model for $T_e$ above 1 mK and on the charge-offset averaged MCMC model for lower temperatures, see the section Integrated Coulomb-blockade thermometry and Supplementary Note 1. The inferred $T_e$ values are listed in the legends. See the Data Availability section for raw data.

demonstrates the importance of a cold electronic environment provided by off-chip refrigeration[27]. Furthermore, due to the finite heat leaks, on-chip refrigeration is required to effectively cool highly resistive devices, where the thermalization via the leads is inefficient. In addition, we extend the range of primary nanoelectronic thermometry below 1 mK, opening avenues for metrological applications and quantum sensing in the ultra-low electron temperature regime. Cooling down electrons to the microkelvin regime also

paves the way towards investigating the limits of quantum coherence in solid state devices on the macroscopic scale[40].

## Methods

**Mechanical design**. The copper stage is milled from 5N5 purity copper. It is left unannealed to reduce eddy currents. The connection to the heat switch is a single M4 bolt, with both the mating surfaces gold plated. The single heat switch in this design resides in a field compensated region of the main magnet. It is formed with

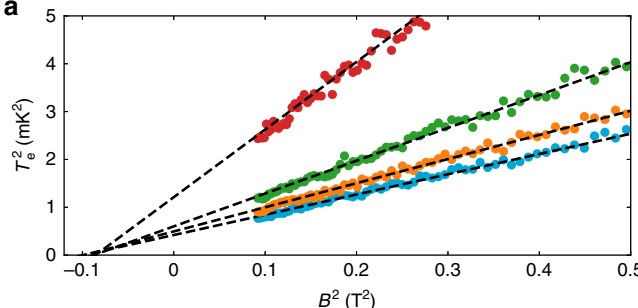

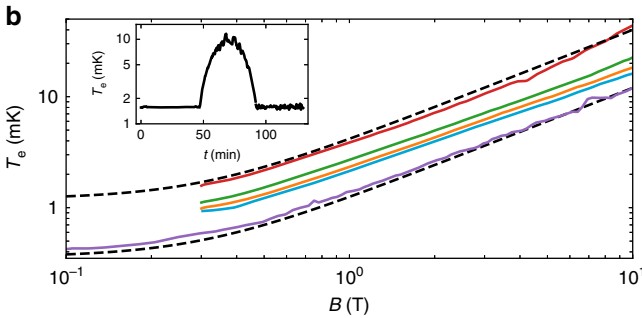

**Fig. 4 Adiabaticity of indium as a nuclear refrigerant for electrons. a** The intercept of the extrapolated $T_e^2(B^2)$ linear regression (dashed lines) with different starting conditions define the internal magnetic field of $|b| = 295 \pm 7$ mT. The experimental data are shown as circles. **b** The $T_e(B)$ plots demonstrate that the internal field limits the obtained electron temperature. Experimental data are shown as solid lines. The dashed lines with constant $T/\sqrt{B^2 + b^2}$ assuming perfect adiabaticity are shown with two different starting conditions. The inset shows a magnetic field ramp $B = 1\,T \rightarrow 8\,T \rightarrow 1\,T$ yielding an adiabatic efficiency of $\eta = 1.00 \pm 0.04$, see the section Internal magnetic field and heat leak calculation. See the Data Availability section for raw data.

a welded lamination of aluminum foil, connecting to copper foil at each end. The aluminum foil layers are mechanically deformed removing continuous paths for superconducting vortices. The switch is then encapsulated in epoxy laced with copper powder for stiffness and filtering. A small superconducting solenoid surrounds the heat switch.

The heat switch shows closing behavior at an applied current of ±90 mA through the solenoid, without measurable dependence on the main field. We observe an abrupt change in both the mixing chamber and CBT temperatures when the solenoid is energized. To be confident of a closed switch, even at high fields, we choose a working current of 140 mA to close the switch. The superconducting lines to the switch quenched when operated above approximately 300 mA. The heat switch current was sourced from a TTi EL301R Power Supply (PSU) running in constant current mode. The copper magnet leads are filtered and thermalised with winding embedded in copper powder epoxy at the 4 K plate before transitioning to superconducting NbTi. The room temperature wiring is wrapped around a ferrite ring for additional filtering. However, when the heat switch power supply is attached, the galvanic isolation to the cryostat from the measurement electronics is broken, so for best noise performance the leads to the PSU are physically unplugged when performing the nuclear cooling experiments.

The molar amount of material in the copper stage is calculated to be 1.7 moles. However, as not all of the material is in the field center, the contribution to the heat capacity and nuclear cooling reduces accordingly. We calculate the effective molar amount using the field profile of the main solenoid along the vertical axes, $B(y)$ and the cross-sectional areas, $A(y)$. The effective molar amount is then

$$n^\star = \frac{1}{V_m} \int \left(\frac{B(y)}{B_{max}}\right)^2 A(y) \mathrm{d}y = 0.32 \text{ mol,} \qquad (1)$$

where $V_m$ is the molar volume and $B_{max}$ is the field maximum.

Similarly, we estimate the effective amount of the indium lines to be 17 mmol with a total amount of 23 mmol per line.

The indium stages are made from 6N 2 mm indium wire rolled out with a small non-ferrous metal rolling pin to flatten it onto a thin epoxy membrane on the copper stage. From there, 4N 1 mm indium wires are stir welded at room temperature to ensure good low-temperature interconnects. Connecting to the

chip, we used 4N 0.7 mm indium wires, which are welded to the 1 mm wires and press fit onto the indium-plated bond pads on the chip. We anneal the indium in situ over 2 days at room temperature using a turbo molecular pump before cooling the experiment down.

The electrical lines for measuring the CBT connect from the mixing chamber to the microkelvin stage through NbTi superconducting lines with the surrounding copper matrix removed using dilute nitric acid.

**Zero bias tracking**. During the magnetic field sweeps and warm-up, we infer the electron temperature based on the zero bias conductance of the CBT. However, time-dependent bias voltage drifts can offset the device, resulting in a systematic error in the measured electron temperature. We note that this error source increases as the temperature is lowered due to the increasing curvature of the conductance dip. To mitigate this effect, we continuously tracked the device conductance in a small voltage bias window centered around the conductance minimum, which was monitored on-the-fly. Note the visible drift in Supplementary Fig. 5. The quoted temperatures are then extracted from the conductance minima of a rolling second-order polynomial fit. This procedure removes the systematic positive temperature error introduced at finite voltage biases, but allows for the temperature tracking during demagnetization.

**Filtering of the electronic lines**. We paid attention to condition the RF environment around the chip, installing RF absorbing materials, in the volumes between plates to reduce the resonant quality factor of the chambers, and covered inner surfaces with an infrared absorbing paint. A set of copper grills are added onto the nuclear cooled stage to damp microwave resonances without increasing eddy current heating in the microkelvin stage.

We used copper powder filters at all stages of the dilution refrigerator. The mixture is 1:3 by mass of copper to epoxy, which also matches the thermal expansion of copper. Voids in the epoxy casting were removed in a vacuum chamber. As the epoxy, we used Loctite Stycast 2850FT with catalyst CAT 24LV, chosen for lower viscosity. The copper powder was supplied by ALDRICH Chemistry, and specified as copper powder(spheroidal), 20 ± 10 μm, 99% purity. We note that we used the same mixture in all stages of the setup.

We used a combined RC and copper powder filter anchored to the mixing chamber plate of the dilution refrigerator. The RC filter component provides a sharp cutoff at 50 kHz. The entire board is covered by copper powder epoxy to prevent microwave leakage. The board accommodates 12 lines in arranged in 6 pairs.

The RC filter components are listed below:
SMD multilayer ceramic capacitors, 50603 [1608 Metric] ±5% C0G/NP0. Parts: C0603C152J5GACTU, C0603C332J5GACTU, C0603C472J5GACTU
SMD Chip Resistor, 0402 [1005 Metric], 1.5 kΩ, ERA2A Series, 50 V, Metal film.
Parts: ERA2AEB751X, ERA2AED152X.

In addition, we further thermalize the experiment with home-made filters mounted inside the slots of the copper nuclear cooled stage. These are made from printed circuit boards with a long meandering trace resulting in an effective continuous element filter. We measure $R = 4.6\,\Omega$ line resistance and a $C \approx 250$ pF capacitance to electrical ground defined by the copper stage. The PCBs have four meandering copper traces on each side. The dual copper layers and the 0.8 mm board thickness slot into 1 mm spaces in the copper stage. The gaps are then filled with copper powder epoxy.

## Data availability

Raw datasets measured and analyzed for this publication are available at the 4TU. ResearchData repository, https://doi.org/10.4121/uuid:ffaeb9fc-9baf-428e-8a33-7e4b451d8f9e (ref. [41]).

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

## Acknowledgements
The authors thank J. Mensingh, O. Benningshof, N. Alberts, R.N. Schouten, and C.R. Lawson for technical assistance. This work has been supported by the Netherlands Organization for Scientific Research (NWO) and Microsoft Corporation Station Q. A.G. acknowledges funding from the European Research Council under the European Union's Horizon 2020 research and innovation programme, grant number 804988. Open access funding was provided by Chalmers University of Technology.

## Author contributions
A.G. designed and supervised the experiments. N.Y. designed and fabricated the CBT devices with integrated indium fins. M.S. designed and fabricated the off-chip nuclear cooling stages and filters. M.S. and N.Y. performed the experiments. N.Y. implemented the numerical CBT array simulation. All authors analyzed the data. The manuscript has been prepared with contributions from all the authors.

## Competing interests
The authors declare no competing interests.
