## [Peer Review File · Nature Communications]

Reviewers' comments:

Reviewer #1 (Remarks to the Author):

I liked this paper. It reports a nice piece of work. It is well and comprehensively written and represents the lowest temperatures to which a CBT thermometer has so far been cooled.

I have one or two minor quibbles. The justification in using nuclear cooling as presented on page 3 is a tad misleading, with the distinction between cooling the nuclei and cooling the metal lost in the loose wording.

In the paragraph at the top of the page we have the statement:

"Exploiting the spin of the nuclei [18, 19], this technique has been utilized to cool bulk metals down to the temperature range of $T \sim 100$ pK [20]."

Well, it hasn't. The experiment of [20] refers to cooling the nuclei alone. In my use of English a "bulk metal" implies the whole system not just the nuclei. That just needs a word added to put right.

In the second paragraph of page 3, I would also challenge the statement:

"however the weak hyperfine interaction results in a decoupling of the electron system at much higher temperatures, ~ 100 μ K."

This is certainly not true. It is one of the great advantages of the Korringa coupling between the electron and nuclear systems that NO such decoupling occurs. The authors unknowingly admit this in the next paragraph by stating the heat transfer equation between electrons and nuclei:

$\dot{Q}_{e-n} = \alpha k_B (T_e/T_n - 1)$. (Excuse the lost formatting!)

This equation shows clearly that (at least in the Curie law region where this is applicable) the ratio of the nuclear and electron temperatures is constant whatever the temperature range or heat leak. (which is why demagnetizing copper can achieve electron temperatures in the single microkelvin range).

Finally, for those not familiar with nuclear cooling, in Fig. 3 panel b, it might be helpful just to explain why plotting against $1/T$ yields a straight line with slope giving the heat leak. I doubt if it is immediately obvious to the general reader.

However, those are minor quibbles, easily sorted, and I would otherwise recommend the publication of the paper as is.

Reviewer #2 (Remarks to the Author):

The manuscript reports on the achievement of a 0.5mK temperature for the electrons in a small circuit. This is a new record low value, by a substantial relative margin of about a factor of 6 to my knowledge. This result was obtained by the combination of on-chip and off-chip (connected leads) adiabatic

cooling using indium. Although none of these technical ingredients taken separately are novelties (on/off-chip cooling and indium coolant), it is shown in this work that their efficient combination gives access to sub-mK electronic temperatures. On the less positive side, this achievement might be difficult to transpose to generic small circuits as it was obtained with a very special kind of electronic device, particularly isolated from the connected leads by its matrix structure with tunnel interconnects. Nevertheless, I find that the step is large enough, even with the safer over-estimation of the temperature, to warrant publication in a high impact journal such as nature communications. The manuscript is well written and the analysis is convincing. My suggestions below are relatively small improvements on the presentation.

- page 3, second to last paragraph: please give numbers when referring to the "large molar heat leaks", including also for the total heat leaks which might be more relevant if originating from the connected leads.
- Fig 1b & second paragraph of page 5: what is the thickness of the epoxy membrane?
- Page 7, second paragraph: It would be easier to have E_C also directly given in Kelvin units, as the temperature threshold to non-universal behavior is a fraction of E_C (1.1mK given by the authors at the bottom of the page).
- Fig 3a,b: Could the authors comment on the origin of the faster temperature increase at the beginning ($t \sim 0-2h$), before the slower rise at $1.7\mu K/h$ sets in.
- Page 10, second paragraph: the authors mention that their heat leak per island (27aW) is lower than the value previously obtained with copper, but this second value is not given. Please add this number. As it is possible that the heat leak comes from the connected leads, it would also be useful to quote the total heat leak.
- Fig S6: The parameters used to calculate the increase in T_e (eq S8) for the fit shown in panel B are not provided. Please add this information.

Reviewer #3 (Remarks to the Author):

The manuscript convincingly demonstrates, for the first time, cooling of the electronic temperature of a nanoelectronic device to below 1 mK. This is a very impressive technical achievement, a tour-de-force, and could open up many possibilities for the future. I heartily recommend publication in Nature Comm.

Before publication, I would however recommend that the authors consider the following points for improvement of the manuscript:

When you start discussing Fig. 2 c, I had a hard time following what you are talking about or what is the relevance of the plot. You need to justify the discussion by clearly explaining that you are discussing the theoretical calibration of the experiment, in the regime where the universal theory is not applicable anymore. Also state clearly what the conclusion of the numerics is (i.e. that random offset charges give error, but of a size that does not prevent from reaching conclusions about the electron temperature). In fact, it would perhaps be more logical to combine Fig. 2c into Fig. 3, as the other plots in Fig 2 refer to experiments in the phonon cooled regime.

Related to the above comment, in the caption of Fig 2 c, use the word "theory" to distinguish it from the experimental plots in Fig 2 a, b. Also, in the main text, be more explicit about the "limit of the universal theory", i.e. explain that it is the temperature above which the universal theory without offset charge effects is valid.

Why was the starting temperature of the nuclear demag experiment 13 mK instead of 7 mK achieved in Fig. 2?

On p. 7 when you first cite the lowest temperature as 421 plus-minus 35 microKelvin, you say you have "calculated it using your MCMC model". If you use the word calculated, the reader might misunderstand that it is "just theory", whereas in reality it is your experimental result. So think of some other way of expressing this, like "with the calibration calculated using.." or similar. In addition, you need to explain here how the error was estimated.

Talking about error: I assume it is the standard deviation of the MC simulation. But you have an underlying assumption that all the junction capacitances and, in particular, resistances, are equal. For the supplement, it would be interesting to try to estimate the additional error caused by a distribution of the parameter values, in particular junction resistances, which are exponentially sensitive functions of the physical barrier parameters (width and height)

Fig. 3: make sure that the reader understands that Fig 3 a is an experimental plot, not just theory

Fig 3 c d and e: are the best fits obtained with average MCMC result? Please be more explicit what the theory curves are. (you just say "calculated curves")

Fig. 4: same, which theory was used for the dashed lines?

Conclusions: Could you comment on two issues: Is indium the only material that could be used? What would be your strategy to cool an arbitrary nanoelectronic device, i.e. make it clear here if thick In pads are a must on every device, or explain how efficient the stage cooling without thick on chip pads would be.

Supplement:

p. 3 define PSU

p. 7 define DeltaG

p. 8 why there is no text for subsection "Zero bias tracking" Text is needed to explain the significance of Fig. S5

p 10 See the comments above about errors due to distribution of values of R. Can you calculate the additional error?

p 12 Fig. S8 You must explain what the different theory curves are

p 13 Fig. S9 same, which theory are the curves

Answers to the Reviewers' comments

We are thankful to all the Reviewers for generally positively evaluating our manuscript and for their suggestions for improving the presentation of our results. We address their input point-by-point below in addition to highlighting the changes in the manuscript.

Reviewer #1 (Remarks to the Author):

I liked this paper. It reports a nice piece of work. It is well and comprehensively written and represents the lowest temperatures to which a CBT thermometer has so far been cooled.

I have one or two minor quibbles. The justification in using nuclear cooling as presented on page 3 is a tad misleading, with the distinction between cooling the nuclei and cooling the metal lost in the loose wording.

In the paragraph at the top of the page we have the statement: "Exploiting the spin of the nuclei [18, 19], this technique has been utilized to cool bulk metals down to the temperature range of $T \sim 100$ pK [20]." Well, it hasn't. The experiment of [20] refers to cooling the nuclei alone. In my use of English a "bulk metal" implies the whole system not just the nuclei. That just needs a word added to put right.

We thank the Referee for this clarification, and reformulated the sentence to refer specifically to the nuclear spin temperature.

In the second paragraph of page 3, I would also challenge the statement: "however the weak hyperfine interaction results in a decoupling of the electron system at much higher temperatures, ~ 100 μ K."

This is certainly not true. It is one of the great advantages of the Korringa coupling between the electron and nuclear systems that NO such decoupling occurs. The authors unknowingly admit this in the next paragraph by stating the heat transfer equation between electrons and nuclei: $\dot{Q}_{e-n} = \alpha \kappa^{-1} B^2 (T_e/T_n - 1)$. (Excuse the lost formatting!) This equation shows clearly that (at least in the Curie law region where this is applicable) the ratio of the nuclear and electron temperatures is constant whatever the temperature range or heat leak. (which is why demagnetizing copper can achieve electron temperatures in the single microkelvin range).

We appreciate that the Reviewer raises this point. Indeed, the heat transfer equation results in a T_e/T_n ratio which does not have an explicit T_n dependence. However, for a given heat leak, T_e/T_n increases with decreasing magnetic field, B , which is in turn proportional to T_n (assuming adiabatic demagnetization). Consequently, we can write $T_e/T_n - 1 \sim 1/T_n^2$, which describes the increasing ratio at low temperatures for a given heat leak. The Reviewer rightfully mentions on the other hand that we misquoted the electron temperatures in copper when referring to bulk stages in this paragraph, which we fix now.

Finally, for those not familiar with nuclear cooling, in Fig. 3 panel b, it might be helpful just to explain why plotting against $1/T$ yields a straight line with slope giving the heat leak. I doubt if it is immediately obvious to the general reader.

Following the Reviewer's advice, we added the derivation of this fit in support of the readers of the article.

However, those are minor quibbles, easily sorted, and I would otherwise recommend the publication of the paper as is.

Reviewer #2 (Remarks to the Author):

The manuscript reports on the achievement of a 0.5mK temperature for the electrons in a small circuit. This is a new record low value, by a substantial relative margin of about a factor of 6 to my knowledge. This result was obtained by the combination of on-chip and off-chip (connected leads) adiabatic cooling using indium. Although none of these technical ingredients taken separately are novelties (on/off-chip cooling and indium coolant), it is shown in this work that their efficient combination gives access to sub-mK electronic temperatures. On the less positive side, this achievement might be difficult to transpose to generic small circuits as it was obtained with a very special kind of electronic device, particularly isolated from the connected leads by its matrix structure with tunnel interconnects. Nevertheless, I find that the step is large enough, even with the safer over-estimation of the temperature, to warrant publication in a high impact journal such as nature communications. The manuscript is well written and the analysis is convincing. My suggestions below are relatively small improvements on the presentation.

- page 3, second to last paragraph: please given numbers when referring to the “large molar heat leaks”, including also for the total heat leaks which might be more relevant if originating from the connected leads.

We have added the extracted molar and total heat leaks for the two experiments referenced in the paragraph. in order to facilitate the comparison of the extracted heat leaks, we also clarify that the temperature minimum and subsequent warmup happened during the magnetic field ramp in contrast with the work of Palma et al (Ref. [26]) and the current manuscript.

- Fig 1b & second paragraph of page 5: what is the thickness of the epoxy membrane?

We now include the nominal thickness 40um in the main text.

- Page 7, second paragraph: It would be easier to have E_C also directly given in Kelvin units, as the temperature threshold to non-universal behavior is a fraction of E_C (1.1mK given by the authors at the bottom of the page).

The Referee rightfully notes that the conversion $E_c=232.6\text{neV}\sim k_B*2.7\text{mK}$ helps the comparison with the limit depicted in Fig.2c. We now include it in the main text.

- Fig 3a,b: Could the authors comment on the origin of the faster temperature increase at the beginning ($t\sim 0\text{-}2\text{h}$), before the slower rise at $1.7\mu\text{K/h}$ sets in.

It is clear that the temperature kink in the beginning of the warm-up cannot be explained by the model that describes the rest of the data using a uniform heat leak and device thermal properties. We can speculate that this feature is caused by a fast warm-up of part of the device, which could be the copper interlayer (270nm thickness) or the aluminum device layer (180 nm). Both lack the large internal magnetic field and hence can demagnetize to a lower end temperature than indium, but also represent only a fraction of the heat capacity on each island. The confirmation and detailed investigation of this effect will require further experiments with different device geometries and more sophisticated thermal models.

- Page 10, second paragraph: the authors mention that their heat leak per island (27aW) is lower than the value previously obtained with copper, but this second value is not given.

Please add this number. As it is possible that the heat leak comes from the connected leads, it would also be useful to quote the total heat leak.

We now include the figure 32aW/island from (Ref. [26]) in the text.

- Fig S6: The parameters used to calculate the increase in T_e (eq S8) for the fit shown in panel B are not provided. Please add this information.

We added the fit parameters including their standard deviation in the caption of Fig. S6.

Reviewer #3 (Remarks to the Author):

The manuscript convincingly demonstrates, for the first time, cooling of the electronic temperature of a nanoelectronic device to below 1 mK. This is a very impressive technical achievement, a tour-de-force, and could open up many possibilities for the future. I heartily recommend publication in Nature Comm.

Before publication, I would however recommend that the authors consider the following points for improvement of the manuscript:

When you start discussing Fig. 2 c, I had a hard time following what you are talking about or what is the relevance of the plot. You need to justify the discussion by clearly explaining that you are discussing the theoretical calibration of the experiment, in the regime where the universal theory is not applicable anymore. Also state clearly what the conclusion of the numerics is (i.e. that random offset charges give error, but of a size that does not prevent from reaching conclusions about the electron temperature). In fact, it would perhaps be more logical to combine Fig. 2c into Fig. 3, as the other plots in Fig 2 refer to experiments in the phonon cooled regime.

We thank the Reviewer for the input on our presentation of our results. We now include the conclusions of the numerics in the main text, that is that the 3σ confidence interval (which we use to define the temperature error bars throughout the manuscript) is less than 10% for the entire temperature range of our experiment with the 35 island times 5 rows device geometry. The placement of Fig. 2c was meant to emphasize that it is part of the calibration procedure.

Related to the above comment, in the caption of Fig 2 c, use the word "theory" to distinguish it from the experimental plots in Fig 2 a, b. Also, in the main text, be more explicit about the "limit of the universal theory", i.e. explain that it is the temperature above which the universal theory without offset charge effects is valid.

We now refer to the blue and orange curves in panel c as *theoretical zero bias calibration curve* and *theoretical maximum electron temperature*, respectively, to make sure that the reader understands this distinction. In the main text, we now explicitly note that in the universal behavior range the offset charges do not influence the device conductance.

Why was the starting temperature of the nuclear demag experiment 13 mK instead of 7 mK achieved in Fig. 2?

The ultimate temperature of 7mK in Fig. 2a was reached with a low applied magnetic field of 45mT, which was required to drive the Al and In layers normal. In contrast, the 13mK starting temperature in Fig. 3a was measured at B=12T. In Fig. S11 of the Supplementary Information, we demonstrate that the thermal relaxation time increases with increasing magnetic field. At B=12T, saturation in temperature is still not reached after the quoted precooling time of 164 hours, which then marks a practical trade-off considering starting temperature and the need of filling the cryostat with cryogenic liquids, which results in a partial warm-up of the device. We finally note

that the high magnetic fields increase the heat load on the dilution refrigerator, resulting in a slightly higher base temperature as well.

To help the reader, we display the respective magnetic fields in both plots.

On p. 7 when you first cite the lowest temperature as 421 plus-minus 35 microKelvin, you say you have "calculated it using your MCMC model". If you use the word calculated, the reader might misunderstand that it is "just theory", whereas in reality it is your experimental result. So think of some other way of expressing this, like "with the calibration calculated using.." or similar. In addition, you need to explain here how the error was estimated.

Following the Reviewer's advice, we now write "lowest measured electron temperature $T_e=421\pm 35\mu\text{K}$ calibrated using our MCMC model". We now also refer to the calculated 3σ confidence interval shown in Fig. 2c and discussed in the previous paragraph to make the error estimate transparent for the reader.

Talking about error: I assume it is the standard deviation of the MC simulation. But you have an underlying assumption that all the junction capacitances and, in particular, resistances, are equal.

For the supplement, it would be interesting to try to estimate the additional error caused by a distribution of the parameter values, in particular junction resistances, which are exponentially sensitive functions of the physical barrier parameters (width and height)

The Referee raises an important question on the reliability of the temperature measurement in the presence of unavoidable variation in tunnel junction resistances and capacitances. This error source has been discussed in detail by Hirvi et al, Journal of Applied Physics 80, 256 (1996), which we added as Ref. [38]. Based on their analysis and the known scattering in resistance of our fabrication process (see Ref. [24]), we estimate a temperature error of approximately 1%. We conclude that this error source does not dominate at our lowest measured temperatures. We finally note that our devices feature a much larger overlap capacitor between the islands than the resistive tunnel junction area, therefore the exact temperature error versus parameter scattering curve is still to be calculated for this specific geometry.

Fig. 3: make sure that the reader understands that Fig 3 a is an experimental plot, not just theory

We now use the expression "nuclear demagnetization experiment" in the caption of Fig. 3a.

Fig 3 c d and e: are the best fits obtained with average MCMC result? Please be more explicit what the theory curves are. (you just say "calculated curves")

We now include in the caption that all curves below 1 mK are the result of the charge offset-averaged MCMC simulation, whereas above the temperature, the calculations are performed by the master equation model as described in Ref. [33] and our Supplementary Information.

Fig. 4: same, which theory was used for the dashed lines?

In the caption of the figure, we write that the dashed line in panel a is a linear regression of T^2 vs B^2 , which yields b^2 as the negative intercept on the horizontal axis. Using this value, we use constant $T/\sqrt{B^2 + b^2}$ ratios to generate the dashed curves in panel b.

Conclusions: Could you comment on two issues: Is indium the only material that could be used? What would be your strategy to cool an arbitrary nanoelectronic device, i.e. make it

clear here if thick In pads are a must on every device, or explain how efficient the stage cooling without thick on chip pads would be.

The Reviewer poses an important point here on the general applicability of the method from the materials and device geometry perspective. In the conclusions paragraph, we now include two requirements important for follow-up experiments: (i) the large α/k ratio and (ii) the magnitude of the effective internal field, b , which together influence the ultimate lowest electron temperature. Based on existing literature and our studies, we now also comment on the necessity of off- and on-chip refrigeration, which we conclude as the required combination for highly resistive devices, such as a Coulomb blockade thermometer investigated in this study.

Supplement:

p. 3 define PSU

p. 7 define DeltaG

We added the respective definitions in the relevant parts of the SI.

p. 8 why there is no text for subsection "Zero bias tracking" Text is needed to explain the significance of Fig. S5

We added a paragraph with a more comprehensive explanation of the requirement of the zero bias tracking in the Supplementary Information.

p 10 See the comments above about errors due to distribution of values of R. Can you calculate the additional error?

We added this consideration and the new Ref. [38] to the main text for the information of the reader.

p 12 Fig. S8 You must explain what the different theory curves are

p 13 Fig. S9 same, which theory are the curves

We thank the Reviewer for pointing out the missing definitions in the captions of these figures, which are now added.

REVIEWERS' COMMENTS:

Reviewer #1 (Remarks to the Author):

I am happy that the revised manuscript addresses satisfactorily all the queries I expressed re the original manuscript.

Reviewer #2 (Remarks to the Author):

The authors have implemented the requested clarifications. I recommend publication of the paper without additional modifications.

Reviewer #3 (Remarks to the Author):

The authors have considered all the comments and improved the manuscript accordingly. It can be published.